# Ultra-processed food consumption and the risk of overweight and obesity in adolescents: A systematic review and meta-analysis

Mekuriaw Nibret Aweke[1]*, Habtamu Wagnew Abuhay[2], Miteku Andualem Limenih[2], Anas Ali Alhur[3], Nebebe Demis Baykemagn[4], Gebrie Getu Alemu[2], Makda Fekadie Tewelgne[4], Tirualem Zeleke Yehuala[4]

1 Department of Human Nutrition, Institute of Public Health, College of Medicine and Health Sciences, University of Gondar, Gondar, Ethiopia, 2 Department of Epidemiology and Biostatistics, Institute of Public Health, College of Medicine and Health Sciences, University of Gondar, Gondar, Ethiopia, 3 College of Public Health, Imam Abdulrahman Bin Faisal University, Dammam, Saudi Arabia, 4 Department of Health Informatics, Institute of Public Health, College of Medicine and Health Sciences, University of Gondar, Gondar, Ethiopia

* mekunib@gmail.com

## Abstract

### Introduction

Overweight and obesity during early life increase the risk of premature morbidity and mortality. Adolescent obesity raises the likelihood of developing cardiovascular risk factors, including prediabetes, type 2 diabetes, dyslipidemia, hypertension, liver disease, and metabolic syndrome. Sedentary lifestyles and unhealthy diets are major contributors, with one of the fastest-growing unhealthy eating patterns being the consumption of ultra-processed foods (UPFs). No systematic review and meta-analysis has specifically examined the association between UPF consumption and overweight/obesity in adolescents.

### Objective

To systematically review and conduct a meta-analysis of available evidence on the association between UPF consumption and overweight or obesity among adolescents.

### Methods

We searched PubMed, ScienceDirect, HINARI, Google, and Google Scholar for primary studies reporting UPF consumption and overweight/obesity outcomes in adolescents, without restrictions on language or study period. Study quality was assessed using the Newcastle–Ottawa Scale. Heterogeneity was evaluated using Cochrane's Q test and the I² statistic. Publication bias and small-study effects were

**Data availability statement:** All relevant data are within the paper and its Supporting Information files.

**Funding:** The author(s) received no specific funding for this work.

**Competing interests:** The authors have declared that no competing interests exist.

**Abbreviations:** BMI: Body Mass Index, CI: Confidence Interval, FFQ: Food Frequency Questionnaire, IOTF: International Obesity Task Force, OR: Odds Ratio, PRISMA: Preferred Reporting Items for Systematic Reviews and Meta-Analyses, RR: Relative Risk, SD: Standard Deviation, UPF: Ultra-Processed Food, WHO: World Health Organization

assessed using Egger's regression test (p < 0.05). A random-effects model estimated pooled associations.

## Results

Twenty-three studies involving 155,000 adolescents were included. Adolescents with higher UPF consumption had 63% greater odds of overweight or obesity compared with those with lower intake (OR = 1.63; 95% CI: 1.36–1.95).

## Conclusion

High UPF consumption is associated with an increased risk of overweight and obesity among adolescents. Public health strategies targeting reduced UPF intake and promotion of healthier diets should be prioritized to prevent adolescent overweight/obesity and associated health risks.

---

## Introduction

The burden of non-communicable diseases (NCDs) such as heart disease, type 2 diabetes, and several types of cancer has increased worldwide [1]. The rise of these NCDs is closely related to metabolic disorders, which include hypertension, insulin resistance, central obesity, high blood sugar and dyslipidemia [2]. The risk of chronic diseases in early life is a concerning issue, as these conditions were previously believed to affect only adults [3]. Adolescents are now increasingly being diagnosed with these NCDs [4]. This early onset not only increases the lifetime risk of such diseases but also imposes a considerable long-term burden on healthcare systems worldwide.

A major contributor to both metabolic syndrome and the development of NCDs is sedentary life style and unhealthy diet [5]. Among the most increasing unhealthy eating pattern is the growing consumption of ultra-processed foods(UPFs) [6]. Ultra-processed foods are industrial products made largely from extracted, modified, or synthetic ingredients, created through multiple processing steps for industrial use [7]. Process foods are typically energy-dense and nutritionally poor, high in added sugars, salt, unhealthy fats, and chemical additives [8]. High consumption of UPFs displaces nutrient-rich whole foods and promotes excessive caloric intake, contributing directly to weight gain and adiposity [9].

Adolescents are particularly vulnerable to higher consumption of UPF because of their growing independence, higher nutritional needs, heavy marketing, easy access to tasty foods, and often inactive lifestyles [10]. During this developmental stage, unhealthy diatary pattern can significantly influence growth, hormonal balance, and long-term metabolic health [11]. Several studies have found that adolescents with higher intake of UPFs are more likely to be overweight or obese [12,13].

In the last few decades, there has been a significant increase in early stage of life overweight and obesity which has now become epidemics [14]. Over a period of 40 years, the number of girls with obesity increased tenfold from 5 million in 1975–50

million in 2016, while the number of boys with obesity increased twelvefold from 6 million in 1975–74 million in 2016 [15]. According to a 2019 World Obesity Federation estimation, 206 million children and adolescents between the ages of 5 and 19 would be obese in 2025, and 254 million in 2030 [16].

Overweigh and obesity during early life impart significant risks of premature morbidity and mortality, which may be evident before the age of 30 [17]. Adolescent obesity increase the risk of developing cardiovascular risk factors that include prediabetes, type 2 diabetes, high cholesterol levels, Hypertension, liver diseases, and metabolic syndrome [18]. Additionally, adolescents with obesity can suffer from psychological issues such as depression, anxiety, poor self-esteem, body image and peer relationships, and eating disorders [18,19]. Furthermore, adolescent obesity is associated with a heightened risk of early puberty [20], menstrual irregularities in girls [21], and sleep disorders such as obstructive sleep apnea and linked to notable social disadvantages [22,23].

Overweight and obesity in adolescents are influenced by a range of factors, including sociodemographic and biological determinants, as well as lifestyle behaviors. Studies showed that physical inactivity, frequent consumption of unhealthy or processed foods, prolonged sedentary behavior, and extended periods of screen time are the modifiable risk factors associated with adolescent obesity and overweight [24–26].

Systematic review and meta-analysis on the association of UPFs consumption and the risk of overweight and obesity was done on adult population group. However there is no systematic review and meta-analysis on the link of UPF consumption and overweight/obesity during adolescence. Findings from individual studies on the relationship between UPF consumption and overweight or obesity are inconsistent. For instance multiple studies done in Indonesia, Iraq, Bangladesh and China have reported a positive association suggesting that higher UPF intake increases the risk of overweight and obesity in this age group [27–29]. In contrast, a study conducted in Qatar revealed a strong inverse relationship, indicating that higher consumption of unhealthy foods was linked to a reduced likelihood of being overweight or obese during adolescence [30].

These inconsistency results highlight the need for a systematic review and meta-analysis to comprehensively synthesize the available evidence and clarify the overall strength and direction of the association. This study aims to bring together existing research to better understand how eating ultra-processed foods is linked to overweight and obesity in adolescents. The goal is to provide stronger evidence that can help shape public health recommendations and support efforts to reduce overweight and obesity in young people.

## Methods

### Study protocol and registration

This systematic review and meta-analysis was conducted in accordance with the Preferred Reporting Items for Systematic Reviews and Meta-Analyses (PRISMA) guidelines [31]. The protocol for this review was prospectively registered in the International Prospective Register of Systematic Reviews (PROSPERO) (Registration No: CRD420251121568).

### Search strategy

A comprehensive literature search was conducted across major databases including PubMed, HINARI, Science direct, google scholar and google to retrieve primary observational studies reporting on the association between UPF consumption and the risk of overweight or obesity among adolescents. The keywords and MeSH terms used in the search included: ("ultra processed" OR "ultra processed" OR "ultra-processed" OR "processed food" OR "industrialized" OR "fastfood" OR "fast food" OR "fast-food" OR "junk food" OR "carbonated beverage") AND ("adolescents" OR "teen" OR "teenager" OR "teenagers" OR "youth" OR "young people") AND ("obesity" OR "overweight" OR "BMI" OR "Body Mass Index"). Database searches were conducted to retrieve articles published up to July 21, 2025.. The full search strategies for each database are provided in S1 File.

## Study selection

All retrieved articles were imported into EndNote 21 (Clarivate, Philadelphia, PA, USA) reference manager to remove duplicates. Titles and abstracts were screened for relevance based on predefined eligibility criteria by two investigators (MNA. and NDB.). Full texts of potentially eligible studies were then reviewed in detail. No restrictions were applied on language or year of publication and all relevant studies published up to July, 2025 were included.

## Inclusion and exclusion criteria

We included studies that met the following criteria: (1) published or unpublished full-text articles; (2) conducted among adolescents aged 10–19 years; (3) investigated the association between UPF consumption and overweight or obesity, using defined criteria for both exposure and outcome and (3), primary observational studies(cross-sectional, cohort, or case-control studies).

Studies were excluded if they: (1) lacked sufficient data to calculate effect estimates (e.g., odds ratios or risk ratios); (2) employed experimental or qualitative study designs (e.g., clinical trials, case series, case reports); (3) were not conducted exclusively among adolescents or did not provide stratified results for this age group; or (4) included mixed samples of adolescents, children, and/or adults without separate reporting for adolescents.

Overweight and obesity were diagnosed using standardized growth references, including the WHO Growth Reference (BMI-for-age z-score > +1 SD for overweight and > +2 SD for obesity) [32], the CDC Growth Charts (85th–94th percentile for overweight and ≥ 95th percentile for obesity), and the International Obesity Task Force (IOTF) age- and sex-specific BMI cut-offs corresponding to adult BMI thresholds of 25 kg/m² for overweight and 30 kg/m² for obesity [33]. In addition abdominal obesity was assessed using waist circumference, defined as ≥ 90th percentile for age and sex [34] (Table 1).

## Outcomes of interest

The primary outcome was overweight and/or obesity in adolescents. It is defined by each included study according to internationally recognized criteria. Overweight and obesity were identified based on body mass index (BMI)-for-age z-scores, using cut-off points recommended by the World Health Organization (WHO) (overweight: > +1 SD; obesity: > +2 SD), International Obesity Task Force (IOTF) age- and sex-specific cutoffs, or equivalent national reference standards. Any available categorical (overweight/obese vs. normal weight) were extracted to assess the pooled effect size of the associations.

## Assessment of the quality of the individual studies

Three independent reviewers (GGA. MFT. and TZY.) assessed the methodological quality of the included studies using the Newcastle-Ottawa Scale (NOS) modified for cross-sectional and cohort designs. This tool evaluates selection of

**Table 1. Criteria for Inclusion of Quantitative Studies Based on the Population, Exposure, Comparator, Outcome (PECO) Framework.**

| Criteria | Details |
|---|---|
| Population (P) | Adolescents (10–19 years old) from any geographic region; excludes studies involving mixed populations of adolescents, children, and/or adults without stratified results for adolescents |
| Exposure (E) | Consumption of ultra-processed foods (UPFs) |
| Comparator (C) | Low or no consumption of UPFs, or consumption of minimally processed/unprocessed foods |
| Outcome (O) | Risk of overweight and obesity, assessed by body mass index (BMI), BMI-for-age z-scores, BMI percentiles, or waist circumference |

participants, comparability of study groups, outcome ascertainment, and statistical analysis methods [35]. Studies were evaluated on a 0–10 scale and classified as "Very Good" (9–10), "Good" (7–8), "Satisfactory" (5–6), or "Unsatisfactory" (0–4). Any differences in scoring were resolved through consensus among the reviewers.

### Data extraction

Data extraction was performed independently by three reviewers (HWA. MAL. and MNA.) using a standardized Excel spreadsheet. The following information was extracted from each study: author, publication year, study design, country, study setting, publication type, sampling method, age group of participants, anthropometric measurements used to assess obesity/overweight, method of obesity/overweight measurement, types of ultra-processed foods consumed, response rate, actual sample size, confounders, and odds ratios (OR) or relative risks (RR) (S2 File). Any disagreements in data extraction were resolved by discussion and consensus.

### Statistical analysis

Extracted data were compiled in Microsoft Excel and analyzed using STATA/MP version 17.0 (Stata Corp, College Station, TX, USA). A random-effects meta-analysis using the DerSimonian–Laird method was conducted to calculate pooled effect estimates (odds ratios) for the association between UPF consumption and overweight/obesity in adolescents. Heterogeneity among studies was assessed using the Cochran Q-test and quantified using the I² statistic [36]. Values of <50%, 50–75%, and >75% were interpreted as low, moderate, and high heterogeneity, respectively [37].

For the meta-analysis, we extracted and pooled multivariable-adjusted odds ratios (ORs) whenever reported; in cases where adjusted estimates were unavailable, crude ORs were used. We conducted sensitivity analyses to determine whether any individual study disproportionately influenced the overall findings on ultra-processed food consumption and the risk of overweight and obesity in adolescents. Publication bias was evaluated both visually, using funnel plots, and statistically with Egger's regression test where a p-value below 0.05 suggested possible bias from smaller studies. We applied the trim-and-fill method to adjust the pooled estimates. All results are presented with 95% confidence intervals.

## Results

### Selection and identification of studies

An initial total of 2,100 records was identified and 175 duplicates were removed. From the remaing 1925 studies 1,813 were excluded based on title review, and a further 34 were excluded following abstract screening. The remaining 77 articles were assessed for eligibility through full-text review, resulting in the inclusion of 23 studies in the systematic review and meta-analysis. A flow diagram of study selection is shown in Fig 1.

### Characteristics of included studies

The current systematic review and meta-analysis included 23 studies [27–30, 38–56] involving a total of 155, 000 adolescents. The meta-analysis included observational research designs cross-sectional, case-control, and cohort conducted between 2008 and 2025. The studies included in this review were conducted in different countries with the largest proportion originating from Indonesia [27,48,54,56] with four studies, followed by Bangladesh [29,44,49] and India [40,46,51] with three studies each. In addition two studies were from Brazil [43,45] while China [53], Ethiopia [52], Greece [50], Iraq [28], Jamaica [47], Mexico [38], Poland [42], Qatar [30], South Korea [41], United States [39] and Vietnam [55] each contributed one study.

The largest sample size was from a study conducted in China with 97,536 adolescents, followed by South Korea with 20,497 adolescents, while the smallest sample size was from Indonesia involved in 76 adolescents. The majority of the studies were cross-sectional in design [27–30,39–43,45–47,49–56], with a smaller number employing case-control

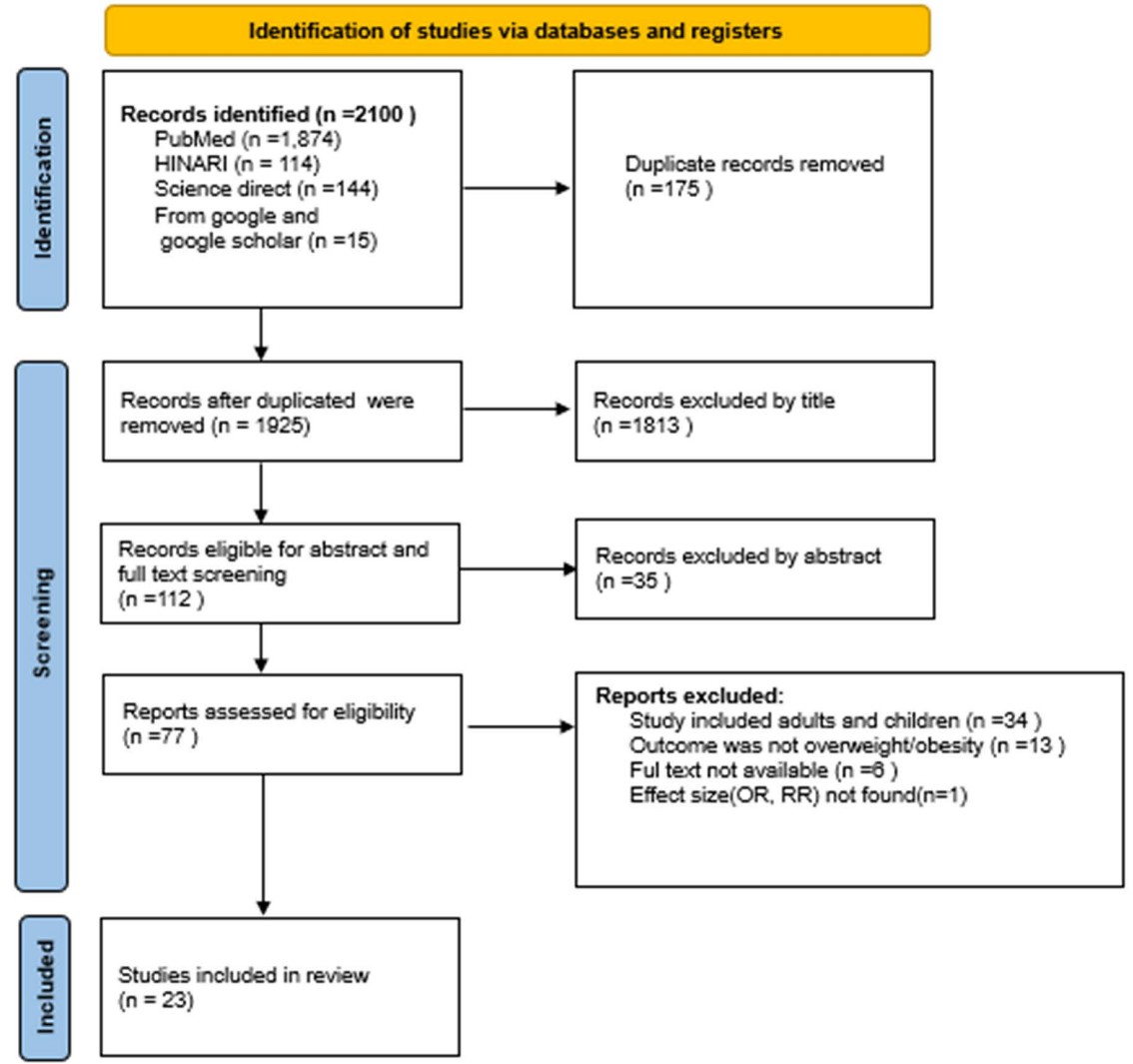

**Fig 1. PRISMA flow diagram for the systematic review and meta-analysis on the association between ultra-processed food (UPF) consumption and obesity among adolescents.**

methods [44,48] and one study utilizing a combined follow-up and cross-sectional approach [38] (Table 2). All included studies were peer reviewed and high methodological quality, with quality scores ranging from six to eight (S3 File).

Among the 23 studies that reported their dietary assessment methods, 19 primarily used food frequency questionnaires (FFQs), while two studies employed 24-hour dietary recalls or food records methods. These studies consistently reported high intake of ultra processed foods such as pizza, burgers, sugary drinks, fried snacks, processed meats, and sweets items typically high in calories, added sugars, unhealthy fats, and low in nutritional quality. The studies assessed overweight and obesity primarily using BMI-for-age indices and Z-scores based on WHO, CDC, or IOTF growth references (Table 3).

## Overall association of ultra-processed foods consumption and obesity/overweight

The meta-analysis revealed a significant positive association between consumption of UPFs and the likelihood of overweight or obesity with a pooled effect size of 1.63 (95% CI: 1.36, 1.95). Adolescents with higher consumption of UPFs

**Table 2. Characteristics of included studies on ultra-processed food consumption and adolescent obesity.**

| S.no | Author | Year | Study design | Country | Study setting | Publication type | Sampling method | Age group | sample size | OR(RR) |
|---|---|---|---|---|---|---|---|---|---|---|
| 1 | Alam et al. | 2018 | case-control | Bangladesh | School-based | Peer-reviewed | Purposive sampling | 10–16 | 240 | 3.05 |
| 2 | B. Saleh and E. Ma´ala | 2015 | Cross-sectional | Iraq | School based | Peer-reviewed | Purposive | 10_19 | 1254 | 1.27 |
| 3 | Banik et al. | 2020 | Cross-sectional | Bangladesh | School Based | Peer-reviewed | Multi-stage random sampling | 10_19 | 518 | 2.71 |
| 4 | Borges et al. | 2018 | Cross-sectional | Brazil | Community-based | Peer-reviewed | Multi-stage cluster sampling | 10–18 | 6784 | 1.55 |
| 5 | Chernet Elias et al. | 2022 | Cross-sectional | Ethiopia | Community-based | Peer-reviewed | multistage | 10_19 | 546 | 2.98 |
| 6 | Denova-Gutiérrez E. et al. | 2008 | Cross-sectional/ Follow-up/ | Mexico | Community-based | Peer-reviewed | convenience | 10_19 | 1055 | 1.55 |
| 7 | Francis DK et al. | 2009 | Cross-sectional | Jamaica | Community-based | Peer-reviewed | Multi-stage random sampling | 15–19 | 1317 | 1.52 |
| 8 | Goyal et al. | 2009 | Cross-sectional | India | school based | Peer-reviewed | multistage | 12_18 | 5664 | 1.54 |
| 9 | Kerkadi et al | 2019 | Cross-sectional | Qatar | School Based | Peer-reviewed | multistage | 14 - 18 | 1161 | 0.7 |
| 10 | Khan et al. | 2019 | Cross-sectional | Bangladesh | school based | Peer-reviewed | clustered sampling technique | 11-17 | 2980 | 2.35 |
| 11 | M. Anitha Rani | 2013 | Cross-sectional | India | Schools | Peer-reviewed | Random sampling | 10_19 | 1842 | 1.905 |
| 12 | Makri et al. | 2022 | Cross-sectional | Greece | school based | Peer-reviewed | multistage | 11, 13, 15 | 3816 | 1.2 |
| 13 | Neri, D. et al. | 2022 | Cross-sectional | USA | Community-based | Peer-reviewed | Multistage | 12–19 | 3587 | 1.07 |
| 14 | Nguyen et al. | 2021 | Cross-sectional | Vietnam | School Based | Peer-reviewed | Random sampling | 11-14 | 2660 | 0.996 |
| 15 | Putra and Santoso | 2024 | Cross-sectional | Indonesia | School Based | Peer-reviewed | Random sampling | 13-18 | 350 | 2.87 |
| 16 | Ra, J. S. & Huyen, D. T. T | 2024 | Cross-sectional | South Korea | School-based | Peer-reviewed | Stratified two-stage cluster sampling | 13–18 | 20497 | 1.12 |
| 17 | Rifqi et al | 2025 | Cross-sectional | Indonesia | School Based | Peer-reviewed | Random sampling | 12–14 | 261 | 2.32 |
| 18 | Souza SF. et al. | 2022 | Cross-sectional | Brazil | School-based | Peer-reviewed | convenience | 10_17 | 576 | 1.58 |
| 19 | Stanislas and Santoso | 2024 | Cross-sectional | Indonesia | School Based | Peer-reviewed | Random sampling | 10_19 | 350 | 2.87 |
| 20 | Vanitha et al. | 2019 | Cross-sectional | India | School Based | Peer-reviewed | Multi-stage random sampling | 13-15 | 230 | 2.84 |
| 21 | Wuenstel, J. W. et al. | 2015 | Cross-sectional | Poland | School-based | Peer-reviewed | Random sampling | 13-19 | 1700 | 1.62 |
| 22 | Yolanda Patri-chia et al. | 2024 | Case-control | Indonesia | School based | Peer-reviewed | Purposive | 16-18 | 76 | 6.98 |
| 23 | Zhihao Huang et al. | 2024 | Cross-sectional | China | School Based | Peer-reviewed | multistage | 12-15 | 97536 | 1.06 |

Footnotes: OR, odds ratio; RR, relative risk.

have 63% greater odds of being overweight or obese compared to those with lower UPF intake. However, there was considerable heterogeneity across the included studies (I²=97.58%, p<0.001) suggesting notable variability in effect estimates. The pooled effect size was visually represented using a forest plot (Fig 2).

**Table 3. Characteristics and Key Findings of Studies on Ultra-Processed Foods and Adolescent Obesity.**

| S. no | Author | Year | Type of UPF | Exposure Assessment | Outcome Assessment | Result | Conclusion |
|---|---|---|---|---|---|---|---|
| 1 | Alam et al. | 2018 | Fast food, cakes/biscuits, chocolates, ice cream, soft drinks | NOVA classification based on FFQ | BMI CDC growth chart | UPF consumption particularly fast foods and cakes/biscuits, was associated with overweight/obesity in adolescents. | Ultra-processed food consumption, contributes to overweight and obesity among adolescents |
| 2 | B. Saleh and E. Ma´ala | 2015 | Fast foods, sugar-sweetened beverages, sweets, snacks/meals in front of screens | NOVA classification based on FFQ | Overweight and obesity was assessed 85th −94th Percentile =>95th Percentile | There is highly significant association between eat the snacks and fast foods in home with their Body Mass Index | Fast foods increase the odds of overweight and obesity |
| 3 | Banik et al. | 2020 | Fast foods: burgers, pizza, fried chicken, puri, chips, etc. | NOVA classification based on FFQ | Overweight was defined as having a BMI between the 85th and 95th percentile, and obesity was defined as having a BMI in the 95th percentile or higher | Frequent consumers of UPF (OR=2.712, 95% CI=1.592–4.622, p<0.05) were the most important factors of obesity. | This study found a positive association between fast food consumption and obesity. |
| 4 | Borges et al. | 2018 | Fast Food and Snacks (e.g., sweetened beverages, sweets, pizzas, processed meats) | NOVA classification based on Food record | BMI standard deviation (SD) 136 scores were used to classify adolescents' nutritional status, whereas overweight>+1SD, obesity>+2SD | Higher the adherence to the To the Fast-Food Pattern (OR: 1.55 fifth quintile vs first (CI95%=1.12; 246 2.12) for linear trend <0,001), the greater the chances of becoming overweight | The greater the adherence to the Fast-Food patterns, the greater the adolescents' chances of becoming overweight. |
| 5 | Chernet Elias et al. | 2022 | Fast foods: pizza, burgers, chocolate, ice cream, chips, soft drinks | NOVA classification based on FFQ | Overweight: BMI-for-age>+1SD; Obesity: BMI-for-age>+2SD (WHO growth reference | Adolescents who are female, come from high-income families, and eat fast-food frequently are more likely to be overweight or obese | Eat fast food frequently are more likely to be overweight or obese. |
| 6 | Denova-Gutiérrez E. Et al. | 2008 | Sweetened beverages (sbs): colas, flavored sodas, sugar-sweetened flavored waters, diet colas | NOVA classification based on Semi QFFQ | WC≥75th percentile and 85th percentile and lower than the 95th percentile | Subjects consuming 3 daily servings of SB face a higher greater risk of proportionally excess body fat than those who consume less than 1 SB a day | The consumption of SB increases the risk of overweight and/or obesity |
| 7 | Francis DK et al. | 2009 | Fast foods and sweetened beverages: sodas, lemonade, Kool Aid, pastries | NOVA classification based on FFQ | Internal BMI Z-scores from LMS method, WC cut-offs at age 18: 93 cm (males) and 81 cm (females). For <18 yrs, equivalent internal Z-score cut-offs were applied. | High WC was associated with the absence of fruit consumption and overweight with high sweetened beverage consumption. | Overweight occurs frequently among Jamaican 15–19-year-olds and is associated with increased consumption of sweetened beverages |
| 8 | Goyal et al. | 2009 | Bakery items, pizza, burgers, cheese, butter, oily items, chocolates | NOVA classification based on FFQ | He BMI of each child was determined and adjusted for expected BMI at age 18 | Eating habit like junk food, chocolate remarkable effect on prevalence on overweight and obesity | Overweight and obesity were associated with life style such as higher consumption of UPFs. |
| 9 | Kerkadi et al | 2019 | junk food, chocolate | NOVA classification based on FFQ | BMI, waist–height ratio (WHtH) | The consumption of an unhealthy diet, especially sugary snacks, while watching TV may be considered a contributing factor for obesity | The present study revealed significant associations between low frequency of intake of breakfast, sugar-sweetened beverages, fast food, fries and sweets and obesity, |

*(Continued)*

| S. no | Author | Year | Type of UPF | Exposure Assessment | Outcome Assessment | Result | Conclusion |
|---|---|---|---|---|---|---|---|
| 10 | Khan et al. | 2019 | Fast foods, low fruit/vegetable intake | NOVA classification based on FFQ | Overweight: BMI > +1SD Z-score (≈25 kg/m² at 19 yrs); Obese: BMI > +2SD Z-score (≈30 kg/m² at 19 yrs), per WHO. | High soft drink Consumption and high fast food consumption were vital influencing factors for being overweight or obese among 16 adolescents across both sexes | Higher consumption of soft drink and fast food were associated with overweight and obesity |
| 11 | M. Anitha Rani | 2013 | Fast food items like cakes, burgers, pizza, noodles, puffs, and fried chaat snacks | Not specified | Overweight (≥85th and <95th%), and obese (≥95th%), according to the guidelines from the CDC | Intake of fast-food increase overwieght and obesity | Higher consumption of fast food is associated with overweight and obesity |
| 12 | Makri et al. | 2022 | Fast foods, sugar-sweetened beverages, sweets, snacks/meals in front of screens | NOVA classification based on FFQ | BMI Z-scores from 2007 WHO growth charts: Underweight <−2SD; Overweight > +1SD to ≤ +2SD; Obesity > +2SD (age- and sex-specific). | Daily consumption of sweets compared to eating sweets less than once per week was associated with being over-weight in the total sample | Consumptions of sweet beverages are associated with increasing overweight and obesity. |
| 13 | Neri, D. Et al. | 2022 | Ultra processed food(not specified) | NOVA classification based on 24 dietary recall | BMI, WC | The highest consumption of ultra-processed food was associated with higher odds of total, abdominal, and visceral overweight/obesity. | Study findings showed the associations between ultra-processed foods and increased adiposity and also with metabolically unhealthy phenotypes of obesity in adolescence. |
| 14 | Nguyen et al. | 2021 | Sugar-sweetened beverages (ssbs): soft drinks, tonic drinks, fruit juices, sweetened milk drinks, iced tea, etc. | NOVA classification based on FFQ | He status of overweight or obesity were defined using International Obesity Task-force 120 (IOTF) sex and age specific BMI cut-offs. | Energy intake from 265 all other sugar-sweetened beverages combined showed slightly higher odds for overweight/obesity 266 but not significant. | No significant association was observed with total energy intake from SSB and overweight and obesity |
| 15 | Putra and Santoso | 2024 | Sugary drinks, packaged snacks, fast food, sweet breakfast cereals, processed meats | NOVA classification based on FFQ | The BMI cut-off recommended by WHO for adolescents | The results showed that high UPF consumption (above the median 4.2 servings/day) significantly increased the risk of obesity in adolescents in Jambi City | High UPF consumption is an independent risk factor for the incidence of obesity in adolescents. |
| 16 | Ra, J. S. & Huyen, D. T. T | 2024 | Sugar-sweetened beverages (ssbs): sodas, sports drinks, other sweetened drinks | NOVA classification based on 7days FFQ | Overweight (≥85th and <95th percentile), and obese (≥95th percentile) according to the screening criteria for growth abnormalities in the 2017 Korean national growth charts | Low and medium consumption of SSBs were associated with increased likelihood of obesity | SSBs consumptions were positively associated with obesity among adolescents. |
| 17 | Rifqi et al | 2025 | Sugar-sweetened beverages (ssbs): sweet drinks, fruit juices, fatty fried foods | NOVA classification based on FFQ & 24-hour dietary recall | Nutritional status assessed by BMI-for-age Z-scores. | Key risk factors for overweight status included sugary drink consumption (OR 2.32), high-fat food intake (OR 1.61). | Sugary drink consumption was associated with obesity among urban residents. |

*(Continued)*

| S. no | Author | Year | Type of UPF | Exposure Assessment | Outcome Assessment | Result | Conclusion |
|---|---|---|---|---|---|---|---|
| 18 | Souza SF. Et al. | 2022 | Sugar-sweetened beverages: Soft/energy drinks, artificial juices Industrialized sweets: Candies, chocolates, cookies, cakes Flours: Industrialized cereal flours Dairy products: Cream cheese, flavored dairy drinks Snacks & fast foods: Chips, pizza, pasta, hot-dogs, fries Sauces & fats: Ketchup, mayonnaise, margarine Processed meats: Sausages, processed meat dishes Instant foods: Instant noodles, ready-made soups | NOVA classification based on monthly FFQ | (BMI/A) was assessed using the curves by sex recommended by the World Health Organization (WHO), The waist circumference was used as an indicator of abdominal obesity, considering the 80th percentile, waist-to-height ratio (WHTR) | ultra processed(prevalence ratio of 1.58; 95%CI: 1.07–2.34) was associated with a higher prevalence of overweight | A higher consumption of processed foods associated with culinary ingredients was related to being overweight, and ultra processed foods with overweight and abdominal obesity. |
| 19 | Stanislas and Santoso | 2024 | High-calorie junk foods: fast food, sugary drinks, packaged snacks, fried foods | NOVA classification based on FFQ | Obesity was defined as BMI ≥ 95th percentile for age and sex based on WHO standards. | Adolescents who frequently consume junk food have a 2.87 times higher risk of becoming obese than those who rarely consume it | Consumption of high-calorie foods (junk food) has a significant role in increasing the risk of obesity in high school adolescents |
| 20 | Vanitha et al. | 2019 | Junk foods: foods made of white flour, fried/salty snacks, carbonated drinks | NOVA classification based on FFQ | Overweight is defined as Body Mass Index (BMI) at or above the 85th percentile or>+1SD and below the 95th percentile | There was significant association with frequency of junk food consumption and BMI variation. | here was significant association with frequency of junk food consumption and BMI variation. |
| 21 | Wuenstel, J. W. Et al. | 2015 | Fruit juices and sweetened beverages (carbonated and non-carbonated) | NOVA classification based on 7days FFQ | BMI cut of point | Daily consumption of sweetened beverages among young people increased the risk of being overweight by more than 60% compared to young people consuming them once a week. | The prevalence of overweight among adolescents was associated with the frequency of sweetened beverage consumption. |
| 22 | Yolanda Patrichia et al. | 2024 | Fast foods: pizza, burgers, chocolate, ice cream, chips, soft drinks | NOVA classification based on FFQ | Obesity was measured using BMI-for-age index (BMI/A) | There is a relationship between junk food consumption patterns and the incidence of adolescent obesity | The prevalence of overweight among adolescents was associated with the frequency of sweetened beverage consumption, but not with the frequency of fruit juice consumption. |

*(Continued)*

**Table 3.** (Continued)

| S. no | Author | Year | Type of UPF | Exposure Assessment | Outcome Assessment | Result | Conclusion |
|---|---|---|---|---|---|---|---|
| 23 | Zhihao Huang et al. | 2024 | Fast foods | NOVA classification based on FFQ | BMI = weight(kg)/height(m)²; overweight/obesity defined by IOTF age- and sex-specific cut-offs | Frequent fast-food consumption was associated with overweight/obesity | This study reported that consumption of SSBs, screen-based sedentary behaviors, and sleep duration might be important targets to prevent adolescent obesity |

The subgroup analysis by study design showed a pooled effect size of 1.55 (95% CI: 1.29, 1.86) for cross-sectional studies 1.55 (95% CI: 1.33–1.81) for cross-sectional follow-up studies, and a markedly higher effect size of 4.71 (95% CI: 2.09–10.58) for case–control studies (Fig 3).

Subgroup analyses by geographic region indicated that the positive association between UPF consumption and overweight/obesity was observed across all regions, though the magnitude and heterogeneity varied. In Africa, evidence from a single study demonstrated a strong association (OR = 2.98, 95% CI 1.88, 4.73). In Asia, the pooled odds ratio demonstrated a significant positive association between UPF consumption and overweight/obesity (OR = 1.73, 95% CI 1.34, 2.23). Similarly, in Europe, the pooled estimate indicated a modest but statistically significant association (OR = 1.31, 95% CI 1.00–,.70). In North America, the pooled analysis also revealed a significant association (OR = 1.33, 95% CI 1.02–1.73). Likewise, evidence from South America showed a significant positive relationship (OR = 1.58, 95% CI 1.07, 2.34) (Fig 4).

Subgroup analyses by sampling approach revealed that studies using non-random sampling showed a statistically significant association between UPF consumption and overweight/obesity (OR = 1.91, 95% CI 1.20–3.04). Similarly, studies employing random sampling also demonstrated a significant association (OR = 1.58, 95% CI 1.29, 1.93). Heterogeneity was present in both subgroups due to the variability in effect sizes across individual studies (Fig 5).

The subgroup analysis based on year of publication showed that the most recent studies (2024–2025) reported the highest pooled odds ratio, with an OR of 2.09 (95% CI: 1.24, 3.51). The findings revealed that the recent rise in ultra-processed food (UPF) consumption is closely linked to the increasing risk of overweight and obesity (Fig 6).

**Publication *Bias*.** Studies were assessed for potential publication bias using funnel plots and Egger's regression test. The Egger's test result (p < 0.05) suggested the presence of small-study effects, consistent with possible publication bias (Table 4). Visual inspection of the funnel plot indicated that unpublished or smaller studies may have had smaller effect sizes compared to published studies (Fig 7) The Trim and Fill method was applied to assess and correct for potential publication bias. This approach estimated the number of studies likely missing to achieve funnel plot symmetry and calculated the effect sizes of these potentially missing studies, providing a bias-adjusted pooled effect size that offers a more conservative and robust estimate of the true effect (Fig 8).

**Sensitivity analysis.** Sensitivity analysis was performed to assess the effect of each study on the pooled estimated association by excluding each study step-by-step from the analysis process and the result showed that excluded studies led to no significant changes in the shared estimation of the association. The results demonstrated that omission of any single study did not materially alter the statistical significance of the association, indicating that the overall finding was robust and not driven by a single influential study. However, as shown in the sensitivity plot, the pooled odds ratios obtained after excluding individual studies were generally lower than the overall estimate, decreasing from 1.62 to approximately 1.46 when each study was omitted in turn (Fig 9).

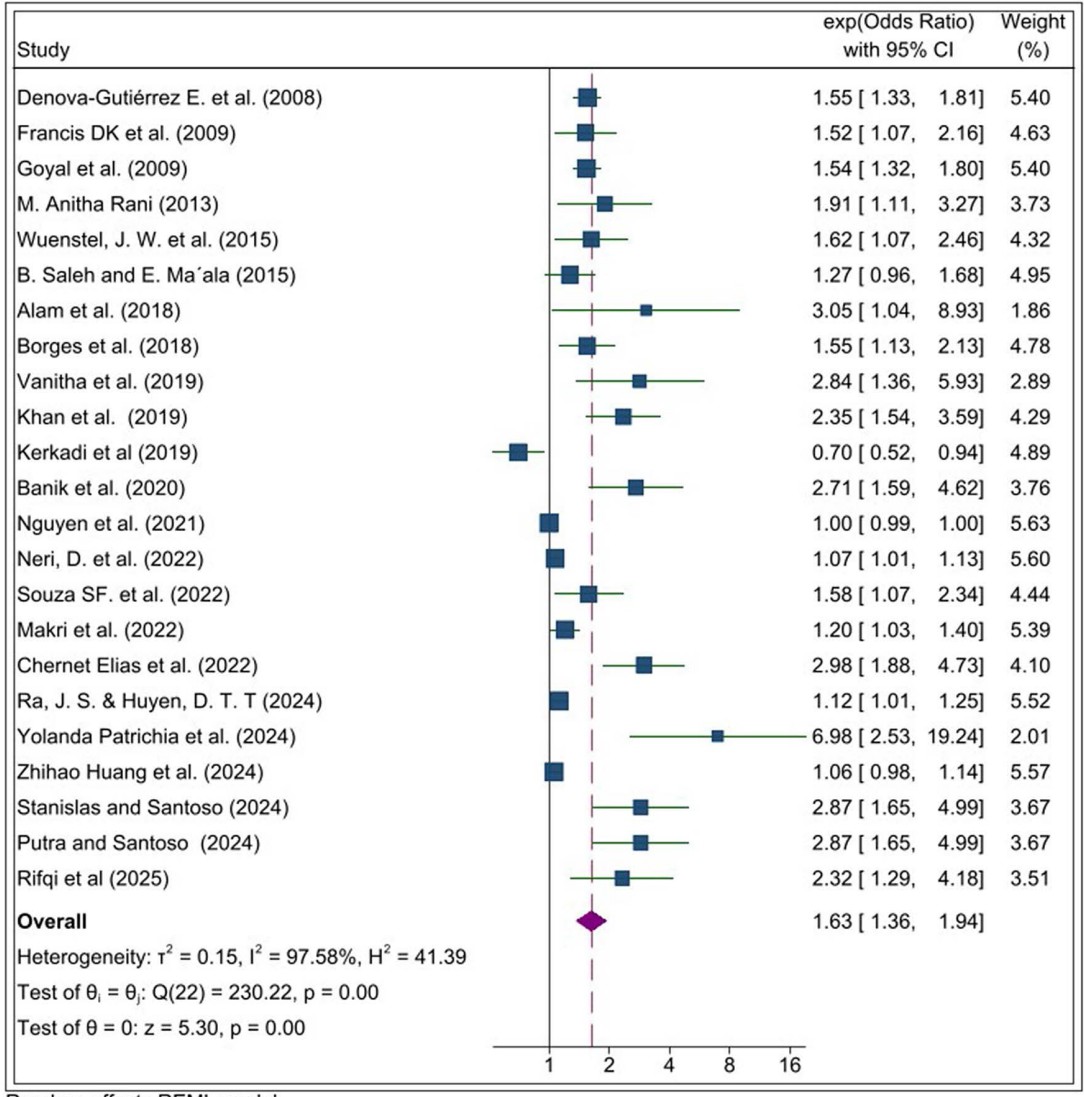

**Fig 2. Forset plot for systematic review and meta-analysis on the association between ultra-processed food (UPF) consumption and obesity among adolescents.**

## Discussion

Overweight and obesity among adolescents are increasing alarmingly and exposing them to various health problems including non-communicable diseases [57]. Unhealthy dietary patterns are the most common cause that increasing the risk of overweight and obesity along with the risk of non-communicable diseases [58]. The aim of this study is to investigate the association between UPF consumption and overweight/obesity among adolescents.

The current systematic review and meta-analysis revealed that there is an association between UPF consumption and the odds of being overweight or obese. According to the findings, higher consumption of UPFs was associated with a 63% increase in the odds of overweight/obesity among adolescents (AOR = 1.63; 95% CI: 1.36–1.95). This

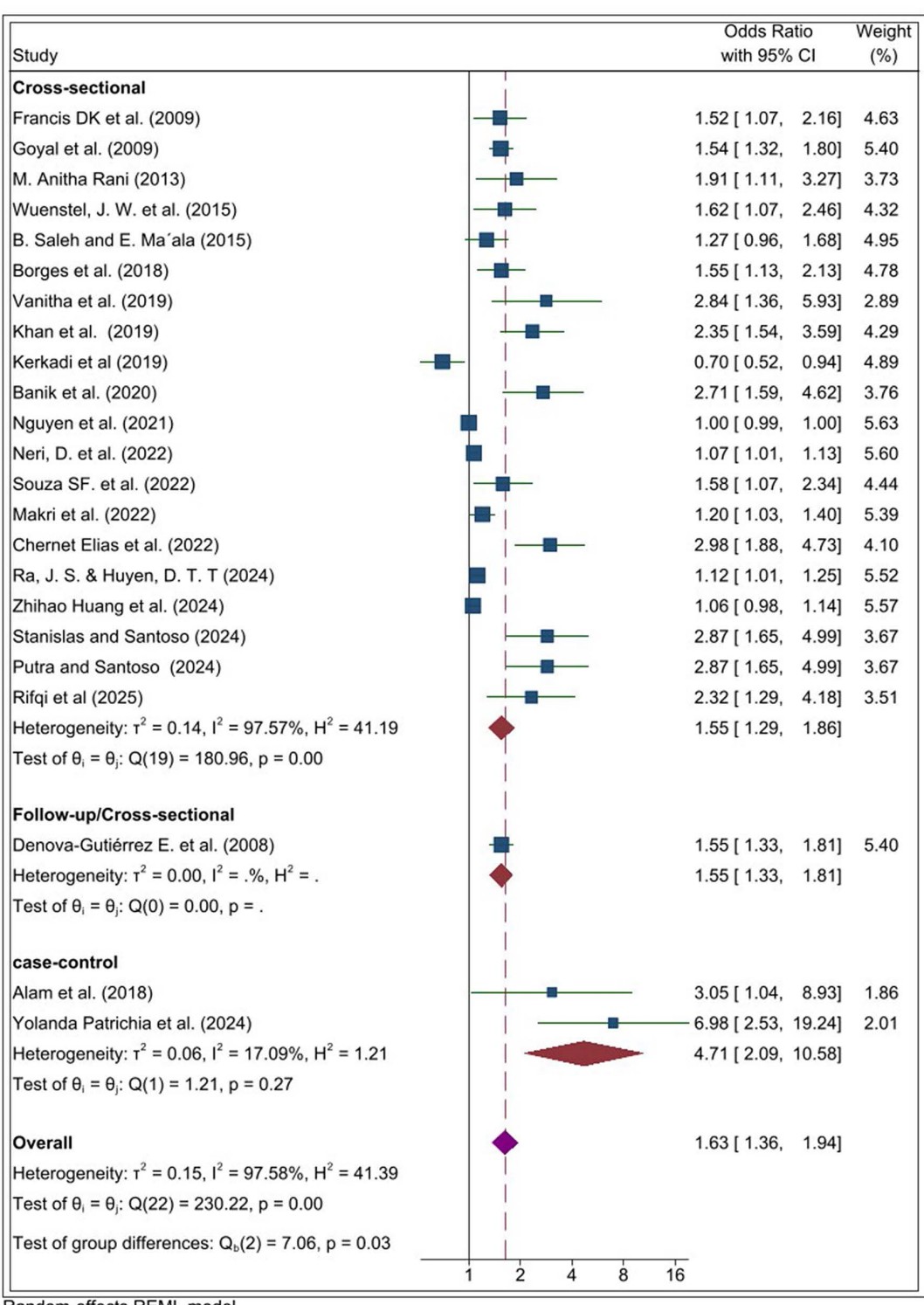

**Fig 3. Subgroup analyses by study designs for the systematic review and meta-analysis on the association between ultra-processed food (UPF) consumption and obesity among adolescents.**

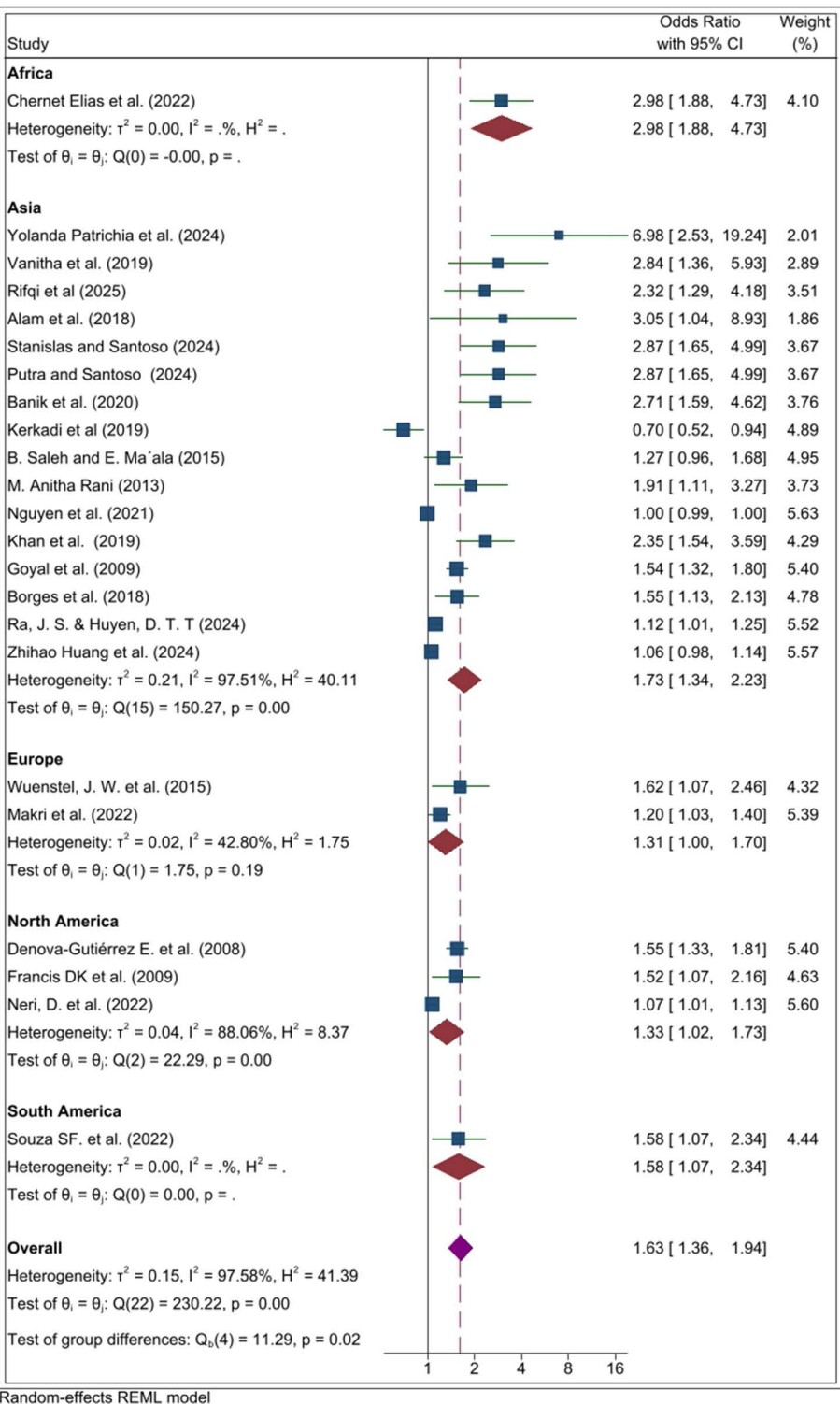

**Fig 4. Subgroup analysis by geographical region in the systematic review and meta-analysis of the association between ultra-processed food (UPF) consumption and obesity among adolescents.**

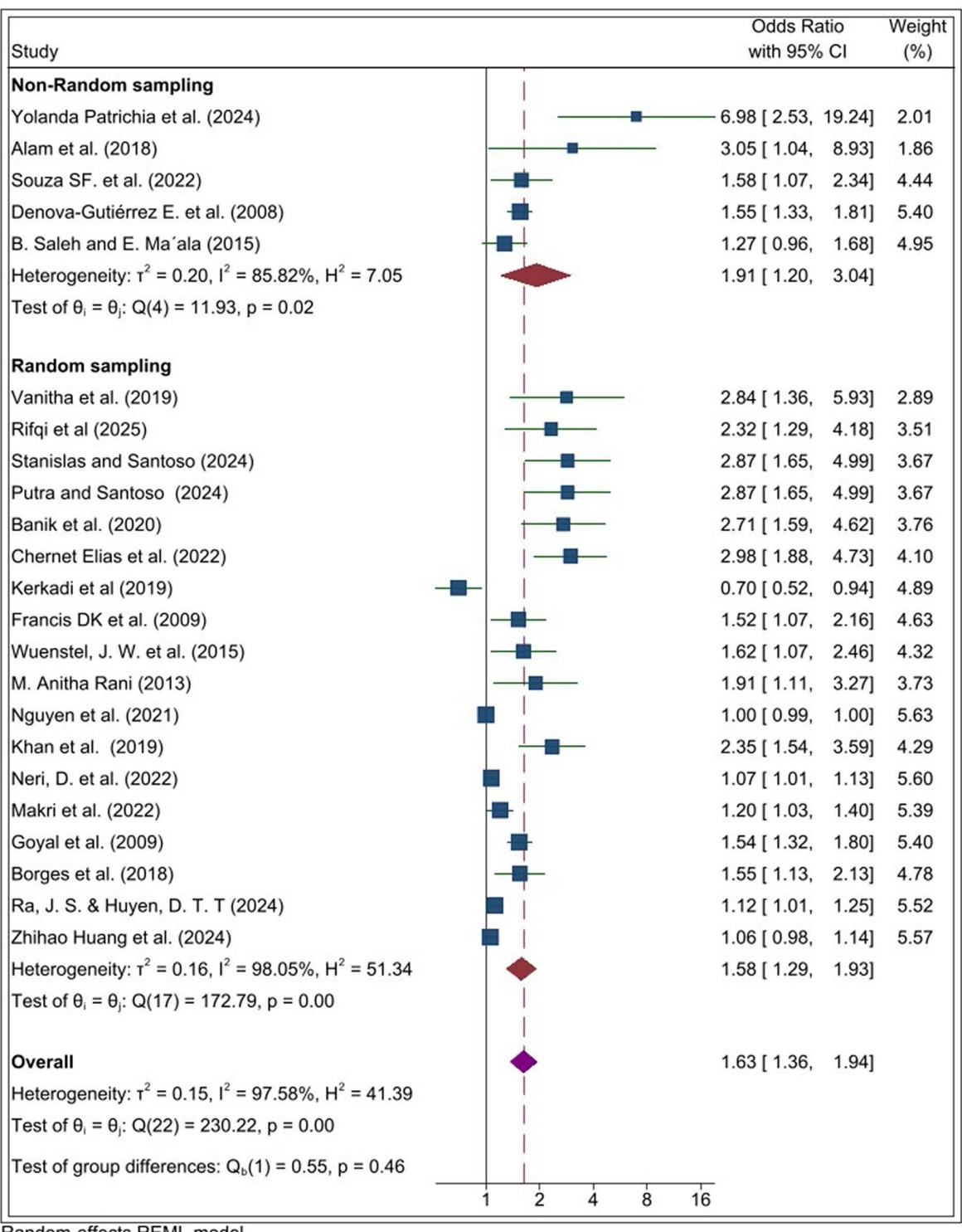

**Fig 5. Subgroup analysis by sampling methods in the systematic review and meta-analysis of the association between ultra-processed food (UPF) consumption and obesity among adolescents.**

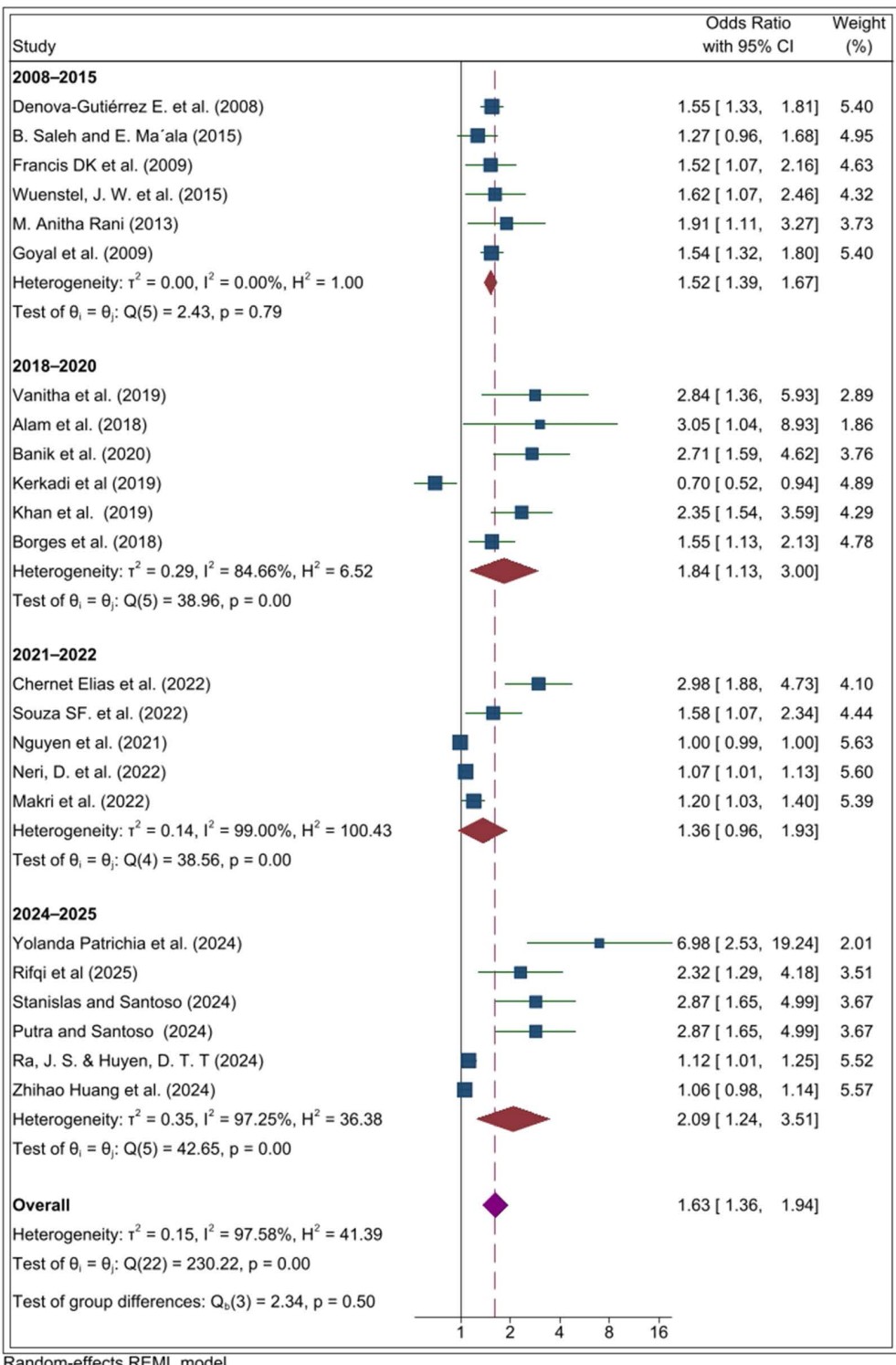

**Fig 6. Subgroup analysis by year of publication in the systematic review and meta-analysis of the association between ultra-processed food (UPF) consumption and obesity among adolescents.**

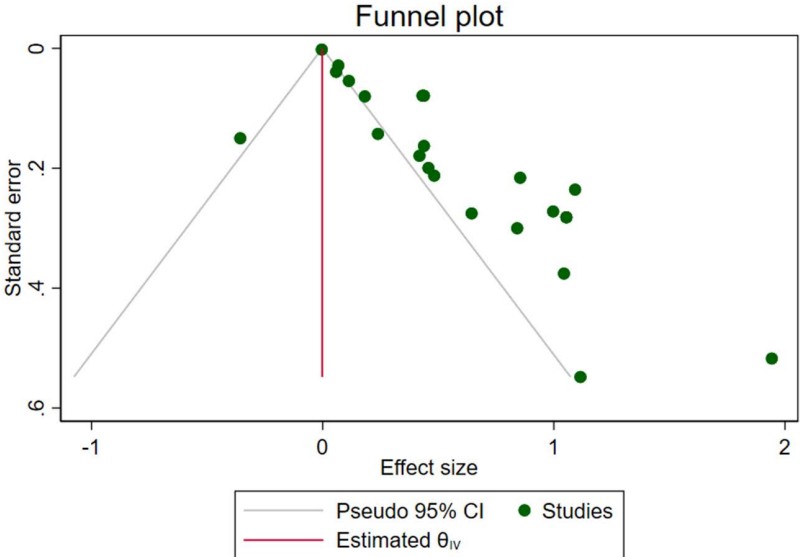

**Fig 7. Funnel plot for publication bias in the systematic review and meta-analysis of the association between ultra-processed food (UPF) consumption and obesity among adolescents.**

**Table 4. Egger's for publication bias in the systematic review and meta-analysis of the association between ultra-processed food (UPF) consumption and obesity among adolescents.**

| Std_Eff | Coefficient | Std. err. | T | P>|t | [95% conf. interval] | |
|---------|-------------|-----------|------|------|--------------|--------------|
| slope | −.0099752 | .0034286 | 1.38 | 0.008 | −.0171052 | −.0028451 |
| bias | 2.882687 | .3510949 | 8.21 | 0.000 | 2.152545 | 3.612829 |

finding aligns with previous research demonstrating that higher consumption of UPFs is associated with an increased risk of overweight and obesity. A systematic review and meta-analysis conducted in 2020 reported a positive association between ultra-processed food intake and excess body weight or obesity with a 2% increase in the odds of overweight and obesity among 10–64 years of the population [59]. Similarly, another systematic review and meta-analysis conducted in 2021 found a significant positive relationship between UPF consumption and the risk of overweight and obesity, reporting a 55% increase in the odds of overweight and a 36% increase in the odds of obesity in the general population [60]. In addition, a systematic review and meta-analysis published in 2024, which assessed the relationship between ultra-processed foods and human health, concluded that UPF consumption is associated with a 32% increase in the odds of obesity [61]. This finding is further supported by umbrella review conducted in 2024 indicated that greater exposure to ultra-processed foods was directly associated with obesity [62]. Although these studies focused on the general population rather than adolescents specifically, their findings are consistent with the patterns observed in our review. While the magnitude of effect differs between adolescents and the general population, the direction of the association remains consistent, strengthening the evidence that higher UPF intake contributes to increased odds of overweight and obesity.

The association between UPF consumption and the increased risk of overweight and obesity can be explained by multidimensional factors. Ultra-processed food are typically energy-dense, nutrient-poor, and high in added sugars, unhealthy fats, and salt, which can promote excessive caloric intake and weight gain [63–65]. Moreover, the high palatability and convenience of UPFs often lead to overconsumption and poor dietary quality, further contributing to obesity development

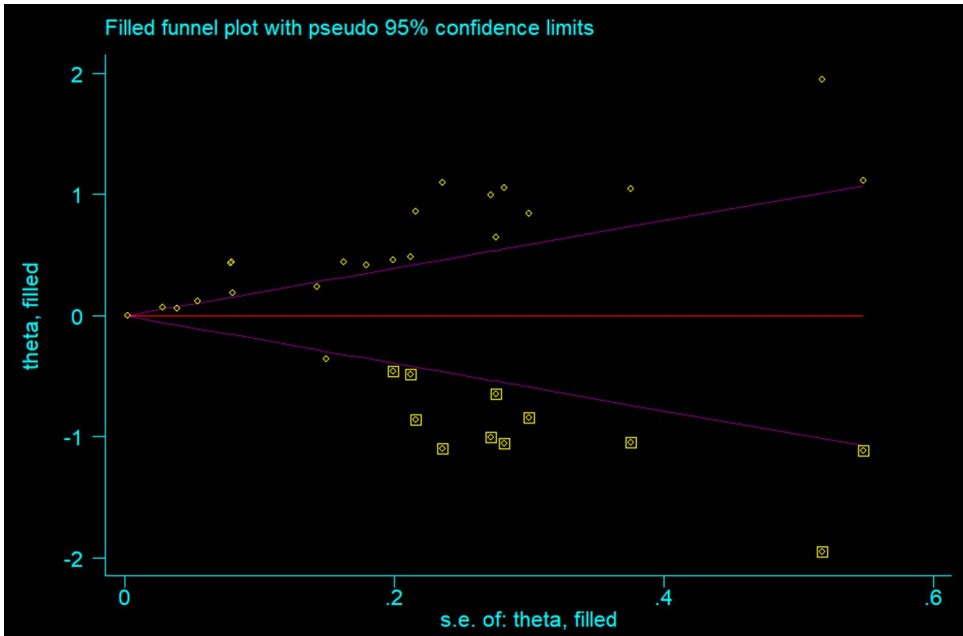

**Fig 8. Trim and fill analysis for publication bias in the systematic review and meta-analysis of the association between ultra-processed food (UPF) consumption and obesity among adolescents.**

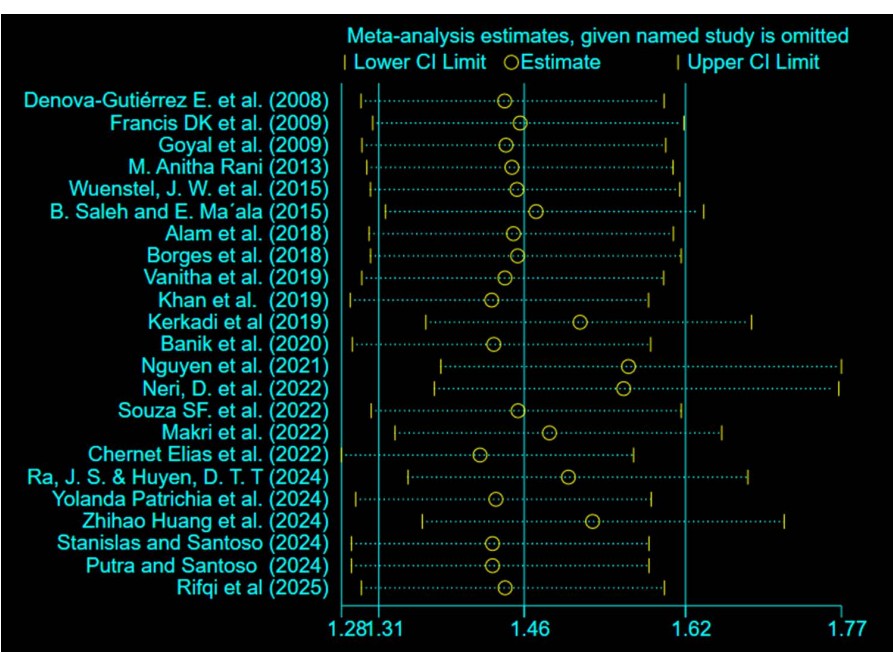

**Fig 9. Sensetivity analysis in the systematic review and meta-analysis of the association between ultra-processed food (UPF) consumption and obesity among adolescents.**

[66]. In addition, the high content of rapidly digestible starches and added sugars in UPFs produces large glycemic loads, triggering sharp postprandial insulin surges [67]. This often results in rebound hypoglycemia, which stimulates hunger and promotes fat storage, further sustaining positive energy balance which increase the risk of overweight and obesity [68,69]. Current evidences showed that UPFs often contain additives such as emulsifiers and artificial sweeteners that can disrupt gut microbiota and promote low-grade inflammation [70,71]. Processing also dismantles the natural food matrix, altering digestion dynamics and appetite regulation by dysregulating endocrine function, insulin signaling, and/or adipocyte function all of these are contributed to overweight and obesity [70,71].

The subgroup analysis indicated that case–control studies reported higher odds of association between UPF consumption and overweight/obesity among adolescents. The finding could be due to the fact that case–control studies, being retrospective and collect exposure data after overweight/obesity onset making adolescents prone to recall bias (overreporting of unhealthy exposures) and selection bias if controls are not truly representative [72]. In contrast, cross-sectional studies measure exposure and outcome simultaneously, which limits causal inference and may underestimate associations due to reverse causation, such as dietary changes following obesity [73]. In addition this may be partly explained by the small number of case–control studies included in this review and meta-analysis, which makes the pooled estimate more susceptible to influenced by to a single outlier with a large odds ratio and affect the overall effect size.

This systematic review and meta-analysis revealed that the most recent studies (2024–2025) showed higher odds of association between UPF consumption and overweight/obesity during adolescent. This is likely related to the increased consumption of ultra-processed foods accompanied by more sedentary lifestyles in recent years [74]. All these together contribute to greater energy imbalance and weight gain. Additionally, improvements in dietary assessment methods and study designs and increase exposure to UPFs in today's food environment may have further contribution the observed associations in these recent studies [75].

## Strengths and limitations of the review

To our knowledge, this is the first systematic review and meta-analysis focusing on the adolescent population, based on a comprehensive literature search across multiple databases with no restrictions on publication year or language. However, limitations include the use of observational studies, restricting causal inference; heterogeneity in dietary assessment methods and BMI reference; and limited representation from low-income settings. The absence of age- and sex-specific reporting in the included studies limited subgroup analyses by age and sex. In addition, publication bias cannot be excluded, potentially inflating observed associations.

## Conclusions and recommendations

The evidence from this study suggests that higher consumption of ultra-processed foods is associated with overweight and obesity among adolescents. This relationship appears consistent across multiple populations and study settings.

Public health strategies should prioritize reducing UPF consumption among adolescents through education, policy interventions, and promotion of minimally processed, nutrient-dense foods. Future research should focus on longitudinal and interventional studies to clarify causal relationships and assess effective strategies for obesity prevention.

## Supporting information

**S1 File. Key words and searching UPF SRMA.**
(DOCX)

**S2 File. Data extraction table.**
(XLSX)

**S3 File. Quality assessment of the included studies.**
(DOCX)

**S4 File. PRISMA checklist.**
(DOCX)

## Acknowledgments

The authors would like to thank all authors of studies included in this systematic review and meta-analysis

## Author contributions

**Conceptualization:** Mekuriaw Nibret Aweke, Habtamu Wagnew Abuhay.

**Formal analysis:** Mekuriaw Nibret Aweke, Miteku Andualem Limenih, Tirualem Zeleke Yehuala.

**Investigation:** Mekuriaw Nibret Aweke, Tirualem Zeleke Yehuala.

**Methodology:** Mekuriaw Nibret Aweke, Anas Ali Alhur, Nebebe Demis Baykemagn, Makda Fekadie Tewelgne, Tirualem Zeleke Yehuala.

**Resources:** Miteku Andualem Limenih.

**Supervision:** Mekuriaw Nibret Aweke, Tirualem Zeleke Yehuala.

**Visualization:** Mekuriaw Nibret Aweke, Tirualem Zeleke Yehuala.

**Writing – original draft:** Mekuriaw Nibret Aweke, Habtamu Wagnew Abuhay, Miteku Andualem Limenih, Nebebe Demis Baykemagn, Gebrie Getu Alemu, Makda Fekadie Tewelgne, Tirualem Zeleke Yehuala.

**Writing – review & editing:** Mekuriaw Nibret Aweke, Habtamu Wagnew Abuhay, Miteku Andualem Limenih, Anas Ali Alhur, Nebebe Demis Baykemagn, Gebrie Getu Alemu, Makda Fekadie Tewelgne, Tirualem Zeleke Yehuala.

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
