## [Decision Letter · Decision Letter 0]

23 Dec 2025

Dear Dr. Aweke,

Thank you for submitting your manuscript to PLOS ONE. After careful consideration, we feel that it has merit but does not fully meet PLOS ONE’s publication criteria as it currently stands. Therefore, we invite you to submit a revised version of the manuscript that addresses the points raised during the review process.

We look forward to receiving your revised manuscript.

Kind regards,

Giulia Squillacioti

Academic Editor

PLOS One

Journal Requirements:

“The authors have not declared a specific grant for this research from any funding agency in the public, commercial or not-for-profit sectors.”

3. We note that you have referenced unpublished data which has currently not yet been accepted for publication. Please remove this from your References and amend this to state in the body of your manuscript: as detailed online in our guide for authors

4. We note you have included a table to which you do not refer in the text of your manuscript. Please ensure that you refer to Tables 1 and 3 in your text; if accepted, production will need this reference to link the reader to the Table.

Reviewers' comments:

Reviewer's Responses to Questions

**Comments to the Author**

1. Is the manuscript technically sound, and do the data support the conclusions?

Reviewer #1: Yes

Reviewer #2: Partly

2. Has the statistical analysis been performed appropriately and rigorously?

Reviewer #1: Yes

Reviewer #2: Yes

3. Have the authors made all data underlying the findings in their manuscript fully available?

Reviewer #1: No

Reviewer #2: Yes

4. Is the manuscript presented in an intelligible fashion and written in standard English?

Reviewer #1: Yes

Reviewer #2: No

Reviewer #1: General comments

I have read the manuscript with interest, and I appreciated the author’s work in addressing such an important topic through a systematic review. This systematic review investigates the presence of evidence on the association between the ultra-processed food consumption and risk of overweight and obesity in adolescents. The use of appropriate methodology and adherence to PRISMA guidelines enhances the credibility of the work and ensures methodological transparency. That said, I think that some revisions can improve the quality and the robustness of the work. Overall, the statistical and methodological parts are very well explained, but some improvements are required. The plots are clear, but a more precise description of the tables and figures could help the readers understand the data. Finally, while the overall structure and content of the manuscript are solid, there are a few areas where the English could be improved for clarity (for instance, there are a few typos that could benefit from a careful reading).

Specific comments

1. Line 65 – missing a “to” in “are particularly vulnerable higher consumption”?

2. Line 91 – typo “there is”

3. Line 92 – typo “during adolescence”, check the entire manuscript.

4. Line 120 – The meaning of this sentence is somewhat unclear; rephrasing it might help improve clarity.

5. Line 123 – when Endnote is cited, the software version and licence should be added.

6. Line 217 – “overall”

7. Line 246 – please, remove the double dot.

8. Line 279-282 – The paragraph currently lacks references, and including some would help situate the discussion within the broader literature.

9. Line 286 – the phrase “increases the likelihood of overweight/obesity in adolescents by 63%” may be misleading, as it could be interpreted as an increase in absolute risk or probability. To improve clarity and avoid potential misinterpretation by non-technical readers, the authors may consider explicitly stating that this figure refers to an increase in odds.

10. Line 289-296 – It would be helpful to expand on the findings from these publications by specifying the populations they investigated and whether the included studies are comparable to those in your review. Additionally, clarifying how these works support your findings—and, where applicable, how your review addresses any limitations they may have—could strengthen your argument. Are their data directly comparable to the percentages reported in your review? Adding a few lines of explanation could enrich the paragraph and help the reader better appreciate the robustness of your work.

11. Please ensure that all acronyms used in the text are clearly defined and explained.

12. Tables: all Tables lack footnotes with abbreviations, which would help the readers to understand the information included.

13. Figures 1-3-4-6 quality needs to be improved.

14. Figure 8 needs a proper description and legend.

15. Figure 4 represents the stratification by year of publication, not by region. Figure 4 needs to be updated.

16. Line 272 – The sensitivity analysis paragraph could be clarified or expanded. While it is correct that excluding individual studies does not alter the statistical significance of the association, the graph shows that most of the pooled estimates obtained after omitting one study are lower than the overall OR (from 1.62 to approximately 1.46). This suggests that some studies contribute to strengthening the overall association. By clarifying this point, methodological transparency would be improved, helping readers better understand the robustness and dynamics of the overall effect.

17. Line 334-335 – I recommend rephrasing this sentence for improved clarity.

Overall, the statistical and methodological parts are very well explained, but some improvements are required. The plots are clear, but a more precise description of the tables and figures could help the readers understand the data.

Reviewer #2: Comments to the authors

The present article aims at filling a void in scientific literature due to the absence of a systematic review and meta-analysis of the evidence regarding the association between the consumption of UPFs and the overweight/obesity in adolescent. The data showed are relevant, especially the pooled estimation derived from the observational studies that represented the majority of the studies. However, the quality of the language, both from a grammatical and syntactical point of view, employed in this paper needs to be inspected and improved to be fully comprehensible and, then, considered for publication.

Methods

Comment 1. The use of Nova classification system to categorise food on the base of level of processing is defined as strength of this study. For the same reason, why was Nova system not considered among the inclusion criteria for the selection of the studies?

Comment 2. Please, add a reference for each criteria used to define overweigh and obesity.

Comment 3. From a methodological point of view this work is solid, but have some limitations. Firstly, the authors should implement sub-group analysis for age, since the age range considered put together adolescents in a different stage of their maturity with likely different eating habits, and for sex, because in this phase energy expenditure and intake starts to diverge between boys and girls. In case it is not possible to conform with this request, it would be appropriate to report the lack of sub-group analysis for age and sex in the limitation of the paper.

Comment 4. I suggest to specify the nature of the pooled measures (odds ratios), univariate or multivariate?

Results

Comment 3. I would add the number of each dietary methods used between parentheses (lines 210-211).

Comment 4. The results showed in the lines 218-222 are repeated twice.

Comment 5. The range 10-19 years is a wide age range that puts together adolescents that might have significantly different dietary behaviours and food intake, especially comparing boys and girls. Consequently, I reckon that additional analyses stratified by age and sex, if it possible, would improve the quality of the evidence.

Discussion

Comment 6. I would add more details of the studies which results were reported from the line 288 to 296, focusing on the peculiarities and differences that these provides compared to the study conducted by the authors of this manuscript. In particular, since the authors stated that this work is the first that assessed the relationship between UPFs consumption and obesity/overweight in adolescents, it should be highlighted that the reviews used in support of their results are based on adult populations.

Comment 7. I would recommend to support your statements about the possible explanations of your result in lines 312-317 with proper references.

Comment 8. A conclusive statement (lines 328-329) is redundant in the discussion; thus, you should relegate your conclusions only to the “conclusion” paragraph of the manuscript. Therefore, I recommend to remove this short paragraph from the discussion. In case the authors would use it as the conclusion paragraph, I recommend to temper what is stated, since it might imply a causal relationship between UPFs consumption and overweight/obesity in adolescent (“…risk of becoming overweight/obese…”).

**Do you want your identity to be public for this peer review?** For information about this choice, including consent withdrawal, please see our Privacy Policy

Reviewer #1: No

Reviewer #2: No

---

## [Author Response · Author response to Decision Letter 1]

25 Jan 2026

25 Jan, 2026

Dear Editors of the Plos One,

We are grateful for the time and expertise devoted to reviewing manuscript PONE-D-25-46416 titled “Ultra-Processed Food Consumption and the Risk of Overweight and Obesity in Adolescents: A Systematic Review and Meta-Analysis”. We sincerely appreciate the reviewers’ constructive feedback, which has helped us improve the scientific clarity and overall quality of the paper.

In response, we have carefully revised the manuscript and provided a detailed, point-by-point reply outlining how each comment has been addressed. Revisions within the manuscript are clearly marked for easy review. We believe these changes strengthen the analysis, interpretation, and presentation of our findings.

Thank you for the opportunity to revise our work.

Warm regards,

Mekuriaw Nibret Aweke

Corresponding Author

1. Editors comment

1. Please submit your revised manuscript

Authers response:

Dear Editor,

Thank you for the opportunity to revise our manuscript based on the reviewers’ comments and suggestions. We have carefully considered each point and revised the manuscript accordingly. The updated version has now been resubmitted for your review. We hope that these revisions adequately address all concerns and have strengthened the overall quality of the manuscript

Authers response:

Dear editor, thank you very much for the reminder to meets the PLOS ONE’S style requirement. We have revised the manuscript in accordance with the journals style requirement including file naming.

3. Please remove any funding-related text from the manuscript and let us know how you would like to update your Funding Statement.

Authers response:

Dear editor, we have revised the manuscript by removing funding information form the main manuscript document.

4. We note that you have referenced unpublished data which has currently not yet been accepted for publication. Please remove this from your References and amend this to state in the body of your manuscript: as detailed online in our guide for authors

Authers response:

Dear editor, thank you very much for your request to remove the unpublished article from the reference list. We have removed the unpublished article from the reference list and we cited with published articles.

5. Please ensure that you refer to Tables 1 and 3 in your text; if accepted, production will need this reference to link the reader to the Table.

Authers response:

Dear editor, thank you very much for your request to refer table 1 and 3 in the text of our manuscript. Accordingly we have corrected by referring table and table 3 with respective for each stetment and now the issue is resolved. Thank you very much again for your suggesions

2. Revewer #1 comments

I have read the manuscript with interest, and I appreciated the author’s work in addressing such an important topic through a systematic review. This systematic review investigates the presence of evidence on the association between the ultra-processed food consumption and risk of overweight and obesity in adolescents.

Response to Reviewer:

Thank you for your positive feedback on the quality relevance and importance of our study. We are very pleased to hear your appreciation, and we value your comments and suggestions, which have helped us improve the manuscript.

1. Line 65 – missing a “to” in “are particularly vulnerable higher consumption”?

Response to Reviewer:

Dear Reviewer, Thank you for your helpful comment. We have add “to” in line 65 stetment and now the statement read as the follows:

“Adolescents are particularly vulnerable to higher consumption of UPF because of their growing independence, higher nutritional needs, heavy marketing, easy access to tasty foods, and often inactive lifestyles.”

2. Line 91 – typo “there is”

Authors response

Dear Reviewer, thank you for very much for your help to correct typo error. We have noticed that the word there was not written correctly. Now we have revised it accordingly thank you again for your valuable comments.

3.Line 92 – typo “during adolescence”, check the entire manuscript.

Authors response

Dear Reviewer, thank you for very much for your help to correct typo error. We have noticed that the phrase “during adolescent” has typo error and now revised it as “during adolescence”. Thank you for your detail reviewing and crucial comments.

4 Line 120 – The meaning of this sentence is somewhat unclear; rephrasing it might help improve clarity.

Authors response:

Dear reviewer thank you very much for your valuable comment to revise the sttement in line 120. We have recognized that the sttement was not clear and now we have corrected it and reads as the following;

“Database searches were conducted to retrieve articles published up to July 21, 2025.”

5 Line 123 – when Endnote is cited, the software version and license should be added.

Authors response:

Dear reviewer thank you very much for your valuable comment to add the software version and license of endnote in line 123. We have revised the sttement by adding the version of endnote.

6 Line 217 – “overall”

Authors response:

Dear Reviewer, we appreciate your careful reading of the manuscript and your valuable comments on each section. Based on your observation, we identified a typographical error (“over all”) and have corrected it to “Overall” in the manuscript.

7 Line 246 – please, remove the double dot.

Authors response:

Dear Reviewer, we have removed the double dot in line 246. Thank you for your observation.

8 Line 279-282 – The paragraph currently lacks references, and including some would help

Authors response:

Dear Reviewer, thank you very much for your suggesions to include reference for line 279-282. We have included the citations for each stetments. Thank you again for your valuable suggestions.

9 Line 286 – the phrase “increases the likelihood of overweight/obesity in adolescents by 63%” may be misleading, as it could be interpreted as an increase in absolute risk or probability. To improve clarity and avoid potential misinterpretation by non-technical readers, the authors may consider explicitly stating that this figure refers to an increase in odds.

Authors response:

Dear Reviewer, thank you for your recommendation to revise the line 286 stement for better clarity for non-technical reads and we have revised the statment and reads as the following:

“According to the findings, higher consumption of UPFs was associated with a 63% increase in the odds of overweight/obesity among adolescents....”

10. Line 289-296 – It would be helpful to expand on the findings from these publications by specifying the populations they investigated and whether the included studies are comparable to those in your review. Additionally, clarifying how these works support your findings—and, where applicable, how your review addresses any limitations they may have—could strengthen your argument. Are their data directly comparable to the percentages reported in your review? Adding a few lines of explanation could enrich the paragraph and help the reader better appreciate the robustness of your work.

Authors response:

Dear Reviewer, we thank the reviewer for this suggestion. We have expanded the Discussion to specify that the cited systematic reviews and meta-analyses focused on the general population, while our study addresses adolescents specifically. We have also revised the stetments with the percentages of odds in each study findings for more clarity. We also clarified how our findings are consistent with previous work that consistently reported the association of ultra processed food consumption and overweight/obesity.

11. Please ensure that all acronyms used in the text are clearly defined and explained.

Authors response:

Dear Reviewer, we thank you for your comment regarding the clear definition and explanation of all acronyms in the text. We identified a typographical error where “MI” was written instead of “BMI,” which has now been corrected, and the acronym is clearly defined in the manuscript. We appreciate your suggestion.

12. Tables: all Tables lack footnotes with abbreviations, which would help the readers to understand the information included.

Authors response:

Dear Reviewer, we thank you for your comment regarding the lack of footnotes in the table. We have added the footnotes with abrivation where abrivations are not explained with the table. Thank you for your valuable comments.

13. Figures 1-3-4-6 quality needs to be improved.

Authors response:

Dear Reviewer, we thank you for your comment suggestion to improve the quality of figure 1, 3, 4, 6. We have improved the quality of figure 3 4 and 6 by redesigning the figure output and we hope now the texts are readable and the quality of the figure is improved. We appreciate your valuable comment and suggesions.

14. Figure 8 needs a proper description and legend.

Authors response:

Dear Reviewer, thank you very much for your suggestion to add proper discription of the figure 8 and its legend. We have revise the stetments to make it more explanatory. We are open to accept additional comments and to revise if any unaddressed issues. Thank you again..

15. Figure 4 represents the stratification by year of publication, not by region. Figure 4 needs to be updated.

Author response

Dear Reviewer, we apologize for the mismatch between the text and the figure. Upon review, we realized that Figure 4 did not represent the geographical subgroup analysis. This has now been corrected, and Figure 4 accurately displays the geographical variation in effect size. We sincerely appreciate your guidance in helping us address this issue.

16. Line 272 – The sensitivity analysis paragraph could be clarified or expanded. While it is correct that excluding individual studies does not alter the statistical significance of the association, the graph shows that most of the pooled estimates obtained after omitting one study are lower than the overall OR (from 1.62 to approximately 1.46). This suggests that some studies contribute to strengthening the overall association. By clarifying this point, methodological transparency would be improved, helping readers better understand the robustness and dynamics of the overall effect.

Author response

Dear Reviewer, we appreciate valuable comments we have revised the sensetivity paragraph explanation to improve the methodological transparency and the revised paragraph reads as the following:

“..However, as shown in the sensitivity plot, the pooled odds ratios obtained after excluding individual studies were generally lower than the overall estimate, decreasing from 1.62 to approximately 1.46 when each study was omitted in turn.”

17. Line 334-335 – I recommend rephrasing this sentence for improved clarity.

Overall, the statistical and methodological parts are very well explained, but some improvements are required. The plots are clear, but a more precise description of the tables and figures could help the readers understand the data.

Author response

Dear Reviewer, we appreciate the thoughtful feedbacks, suggestions and comments. In response, we have revised the sentence in Lines 334–335 for improved clarity. We have also enhanced the descriptions of the tables and figures to offer clearer guidance and improve readers’ understanding of the statistical and methodological results. We are also open to making any revision if unaddressed concerns. Thank ypu again for your valuable comments that helps to enhance the quality of the work.

3. Reviewer 2 comments and responses

The present article aims at filling a void in scientific literature due to the absence of a systematic review and meta-analysis of the evidence regarding the association between the consumption of UPFs and the overweight/obesity in adolescent.

Author response

Dear Reviewer, we are very glad to hear this positive feedback about the gap filling of our work. We have revised all the concerns raised and we hope the manuscript is now improved and fully comprehensible.

Comment 1. The use of Nova classification system to categorise food on the base of level of processing is defined as strength of this study. For the same reason, why was Nova system not considered among the inclusion criteria for the selection of the studies?

Author response

Dear Reviewer, we thank the reviewer for this comment. The NOVA classification was not included as an eligibility criterion to avoid unnecessarily restricting the evidence base, as many relevant studies assessed food processing using alternative definitions. Restricting inclusion only to studies that explicitly applied the NOVA classification would have introduced selection bias and that is why we didn’t use the inclusion criteria.

Comment 2. Please, add a reference for each criteria used to define overweigh and obesity.

Author response

Dear Reviewer, thank you very much for your recommendation to include reference for each criteria define overweight and obesity. We have included the reference for each classification criteria.

Comment 3. From a methodological point of view this work is solid, but have some limitations. Firstly, the authors should implement sub-group analysis for age, since the age range considered put together adolescents in a different stage of their maturity with likely different eating habits, and for sex, because in this phase energy expenditure and intake starts to diverge between boys and girls. In case it is not possible to conform with this request, it would be appropriate to report the lack of sub-group analysis for age and sex in the limitation of the paper.

Author response

Dear Reviewer, we thank the reviewer for this valuable suggestion. Although subgroup analyses by age and sex would be methodologically informative, they were not feasible because most included studies reported findings for mixed age groups and combined sexes, without providing stratified data. We have now explicitly acknowledged the lack of age- and sex-specific subgroup analyses as a limitation in the revised manuscript. Thank you for this important comment.

Comment 4. I suggest to specify the nature of the pooled measures (odds ratios), univariate or multivariate?

Author response

Dear Reviewer, we appreciate for this valuable comment to make specify the nature of the pooled measure(odds ratios). We have specified the nature of the pooled measures and the revised stetement reads as the follows:

“For the meta-analysis, we extracted and pooled multivariable-adjusted odds ratios (ORs) whenever reported; in cases where adjusted estimates were unavailable, crude ORs were used.”

Results

Comment 3. I would add the number of each dietary methods used between parentheses (lines 210-211).

Author response

Dear Reviewer, we appreciate the valuable comment to add the number studies used diatary assessment methods. Based on the comment we have revised the statement and specified the number of studies used types of diatary assessment methods. Thank you again for this crucial suggesions. The revised stetment reads as the following:

“Among the 23 studies that reported their dietary assessment methods, 19 primarily used food frequency questionnaires (FFQs), while two studies employed 24-hour dietary recalls or food records methods.”

Comment 4. The results showed in the lines 218-222 are repeated twice.

Author response

Dear Reviewer, we thank the reviewer for this comment. We carefully re-examined lines 218–222 of the manuscript but were unable to identify duplicated statements. We are open to making correction if the repeated sttements are found in any parts of the documents. Thank you for your detail reading and valuable comments.

Comment 5. The range 10-19 years is a wide age range that puts together adolescents that might have significantly different dietary behaviours and food intake, especially comparing boys and girls. Consequently, I reckon that additional analyses stratified by age and sex, if it possible, would improve the quality of the evidence.

Author response

Dear Reviewer, we thank the reviewer for this important observation. While age- and sex-stratified analyses would strengthen the evidence,

---

## [Decision Letter · Decision Letter 1]

17 Feb 2026

Dear Dr. Aweke,

**Please address the Reviewer #2 suggestions****Please formally justify the inclusion of the new Author in the co-authors list**

plosone@plos.org . A letter that responds to each point raised by the academic editor and reviewer(s). You should upload this letter as a separate file labeled 'Response to Reviewers'.A marked-up copy of your manuscript that highlights changes made to the original version. You should upload this as a separate file labeled 'Revised Manuscript with Track Changes'.An unmarked version of your revised paper without tracked changes. You should upload this as a separate file labeled 'Manuscript'.

We look forward to receiving your revised manuscript.

Kind regards,

Giulia Squillacioti

Academic Editor

PLOS One

Journal Requirements:

Reviewers' comments:

Reviewer's Responses to Questions

**Comments to the Author**

Reviewer #1: All comments have been addressed

Reviewer #2: (No Response)

2. Is the manuscript technically sound, and do the data support the conclusions?

Reviewer #1: Yes

Reviewer #2: Yes

3. Has the statistical analysis been performed appropriately and rigorously?

Reviewer #1: Yes

Reviewer #2: Yes

4. Have the authors made all data underlying the findings in their manuscript fully available?

Reviewer #1: Yes

Reviewer #2: Yes

5. Is the manuscript presented in an intelligible fashion and written in standard English?

Reviewer #1: Yes

Reviewer #2: No

Reviewer #1: The authors have satisfactorily addressed all the questions and comments raised in my previous review, and I believe the manuscript is now suitable for publication. However, I would like to point out that Figure 8 still lacks a legend and the axes are not defined, which makes the figure difficult to interpret. I recommend addressing this minor issue before final acceptance.

Reviewer #2: The manuscript appears to have improved following the reviewers’ suggestions. However, I still have reservations regarding the authors’ response to Comment 1 from Reviewer 2.

In my opinion, the decision to include all classifications available in the literature to categorise UPFs necessitates the provision of additional methodological detail. First, I recommend adding a new table (similar to Table 2) that includes a column describing the method used to classify UPFs in each study included in the review. Furthermore, a similar level of detail should be provided for the outcomes, specifying the methods used to assess overweight and obesity.

The proposed table could include the following columns: S. No., Author, Exposure Assessment, Outcome Assessment, Results, and Conclusion.

Second, if feasible, the authors should consider conducting subgroup analyses based on the different methods used to assess both exposure and outcomes.

Finally, the statement that the use of the Nova system represents a strength of this review (lines 352–355) should be supported strictly by the results obtained. Such a claim should be made cautiously and only if clearly justified by the findings.

In conclusion, in addition to the suggestions mentioned above to improve the scientific quality of the manuscript, I strongly recommend that the authors seek professional English language editing prior to resubmission. The manuscript contains frequent grammatical errors, awkward sentence constructions, and typographical mistakes, which significantly affect its overall clarity.

**Do you want your identity to be public for this peer review?** For information about this choice, including consent withdrawal, please see our Privacy Policy

Reviewer #1: No

Reviewer #2: No

---

## [Author Response · Author response to Decision Letter 2]

19 Feb 2026

18 Feb, 2026

Dear Editors of the Plos One,

We are grateful for the time and expertise devoted to reviewing manuscript PONE-D-25-46416 titled “Ultra-Processed Food Consumption and the Risk of Overweight and Obesity in Adolescents: A Systematic Review and Meta-Analysis”. We sincerely appreciate the reviewers’ constructive feedback, which has helped us improve the scientific clarity and overall quality of the paper.

In response, we have carefully revised the manuscript and provided a detailed, point-by-point reply outlining how each comment has been addressed. Revisions within the manuscript are clearly marked for easy review. We believe these changes strengthen the analysis, interpretation, and presentation of our findings.

Thank you for the opportunity to revise our work.

Warm regards,

Mekuriaw Nibret Aweke

Corresponding Author

1. Editors comment

1. Please address the Reviewer #2 suggestions

Authers response:

Dear Editor,

We are very pleased to have recived the quick feedback from the editor and reviewers after revision. Accordingly we have corrected and revised the manuscript based on the suggesions of reviewer #2. Thank you very much for the opportunity .

2. Please formally justify the inclusion of the new Author in the co-authors list

Authers response:

We respectfully request approval to add Anas Ali Alhur to the list of authors for this manuscript. The Authors had substantial contributions to analysis, interpretation of data critically revising the manuscript for important intellectual content; and final approval of the version to be published.

He also provided critical intellectual revisions to the manuscript that enhanced its scientific rigor and clarity. Anas Ali Alhur has reviewed and approved the final version of the manuscript and agrees to be accountable for all aspects of the work, ensuring the accuracy and integrity of the research. This authorship update does not affect the study’s results, conclusions, or disclosures, and no conflicts of interest arise from this change.

2. Revewer #1 comments

The authors have satisfactorily addressed all the questions and comments raised in my previous review, and I believe the manuscript is now suitable for publication. However, I would like to point out that Figure 8 still lacks a legend and the axes are not defined, which makes the figure difficult to interpret. I recommend addressing this minor issue before final acceptance.

Response to Reviewer:

Dear reviewer thank you very much for your acceptance that we have addressed the questions and comments. In addition we have revised figure to by including legend and defined the axes which will now easy to interpret.

We appreciate your invaluable feedback to enhance the quality of manuscript.

3. Reviewer 2 comments and responses

The manuscript appears to have improved following the reviewers’ suggestions.

Author response

Dear Reviewer, we are very glad to hear this positive feedback about the improvement of the manuscript after reviewers suggesions.

Comment 1. In my opinion, the decision to include all classifications available in the literature to categorise UPFs necessitates the provision of additional methodological detail. First, I recommend adding a new table (similar to Table 2) that includes a column describing the method used to classify UPFs in each study included in the review. Furthermore, a similar level of detail should be provided for the outcomes, specifying the methods used to assess overweight and obesity. The proposed table could include the following columns: S. No., Author, Exposure Assessment, Outcome Assessment, Results, and Conclusion.

Author response

Dear Reviewer, thank you for this valuable comment. We agree that providing additional methodological detail strengthens the clarity and transparency of the review. Accordingly, we revised the manuscript by adding a new table (Table 3), structured similarly to Table 2, which explicitly details the methods used to classify ultra-processed foods in each included study. This table now includes information on the exposure assessment (including UPF classification approaches), outcome assessment methods for overweight and obesity, as well as the corresponding results and conclusions of each study. These revisions enhance comparability across studies and improve the methodological rigor of the review.

Comment 2 Finally, the statement that the use of the Nova system represents a strength of this review (lines 352–355) should be supported strictly by the results obtained. Such a claim should be made cautiously and only if clearly justified by the findings.

Author response

Dear Reviewer, thank you for this important comment. We agree that claims regarding methodological strengths should be made cautiously and be directly supported by the findings. Accordingly, we have removed the statement asserting the use of the NOVA classification as a strength of the review (formerly lines 352–355) to avoid overstating its contribution beyond what is clearly justified by the results.

Comment 3 In conclusion, in addition to the suggestions mentioned above to improve the scientific quality of the manuscript, I strongly recommend that the authors seek professional English language editing prior to resubmission. The manuscript contains frequent grammatical errors, awkward sentence constructions, and typographical mistakes, which significantly affect its overall clarity..

Author response

Dear reviewer, we thank the reviewer for this valuable suggestion. We have revised spelling errors typographical and other mistakes and we hope this revision will improve the updated version of the manusrcipt.

---

## [Editor Report · Decision Letter 2]

27 Feb 2026

Ultra-Processed Food Consumption and the Risk of Overweight and Obesity in Adolescents: A Systematic Review and Meta-Analysis

PONE-D-25-46416R2

Dear Dr. Aweke,

We’re pleased to inform you that your manuscript has been judged scientifically suitable for publication and will be formally accepted for publication once it meets all outstanding technical requirements.

Kind regards,

Giulia Squillacioti

Academic Editor

PLOS One
---

## [Editor Report · Acceptance letter]

PONE-D-25-46416R2

PLOS One

Dear Dr. Aweke,

I'm pleased to inform you that your manuscript has been deemed suitable for publication in PLOS One. Congratulations! Your manuscript is now being handed over to our production team.

Kind regards,

on behalf of

Dr. Giulia Squillacioti

Academic Editor

PLOS One